# Clinical Applicability of Microbiota Sampling in a Subfertile Population: Urine versus Vagina

**DOI:** 10.3390/microorganisms12091789

**Published:** 2024-08-29

**Authors:** Rivka Koedooder, Sam Schoenmakers, Martin Singer, Martine Bos, Linda Poort, Paul Savelkoul, Servaas Morré, Jonathan de Jonge, Dries Budding, Joop Laven

**Affiliations:** 1Division of Reproductive Endocrinology and Infertility, Department of Obstetrics and Gynecology, Erasmus University Medical Center, 3015 GD Rotterdam, The Netherlands; r.koedooder@erasmusmc.nl (R.K.); j.laven@erasmusmc.nl (J.L.); 2Division Obstetrics, Department of Obstetrics and Gynecology, Erasmus University Medical Center, 3015 GD Rotterdam, The Netherlands; 3Department of Medical Microbiology and Infection Control, Amsterdam UMC, Location VUmc, 1081 HV Amsterdam, The Netherlandspaul.savelkoul@mumc.nl (P.S.); 4Tubascan (Spin-Off Company), Department of Medical Microbiology and Infection Control, Amsterdam UMC, Location VUmc, 1081 HV Amsterdam, The Netherlands; 5inBiome, 1098 XG Amsterdam, The Netherlands; martine.bos@inbiome.com (M.B.); linda.poort@inbiome.com (L.P.); dries.budding@inbiome.com (D.B.); 6Department of Medical Microbiology, Infectious Diseases & Infection Prevention, School of Nutrition and Translational Research in Metabolism (NUTRIM), Maastricht University Medical Center, 6229 HX Maastricht, The Netherlands; 7Department of Molecular and Cellular Engineering, Jacob Institute of Biotechnology and Bioengineering, Sam Higginbottom University of Agriculture, Technology and Sciences, Allahabad 211007, India; samorretravel@yahoo.co.uk; 8Institute of Public Health Genomics, Department of Genetics and Cell Biology, Research Institute for Oncology and Reproduction, Faculty of Health, Medicine & Life Sciences, University of Maastricht, 6211 LK Maastricht, The Netherlands; 9ARTPred B.V., 1438 AX Oude Meer, The Netherlands; jonathan@artpred.com

**Keywords:** urine, vagina, microbiota, IS-pro, IVF/IVF-ICSI

## Abstract

The urogenital microbiota is increasingly gaining recognition as a significant contributor to reproductive health. Recent studies suggest that microbiota can serve as predictors for fertility treatment outcomes. Our objective was to investigate the degree of similarity in microbial composition between patient-collected urine and vaginal samples in a subfertile population. We enrolled women of reproductive age (20–44 years) diagnosed with subfertility and requiring in vitro fertilization (IVF) or IVF with intracytoplasmic sperm injection (IVF-ICSI) treatment. They self-collected both mid-stream urine samples and vaginal swabs before commencing the IVF or IVF-ICSI procedure. All samples were analysed using the intergenic spacer profiling (IS-pro) technique, a rapid clinical microbiota analysis tool. The main outcome measures were the degree of similarity of microbial composition between the two different, but simultaneously collected, samples. Our findings revealed a high correlation (R squared of 0.78) in microbiota profiles between paired urine and vaginal samples from individual patients. Nevertheless, the urinary microbiota profiles contained fewer species compared to the vaginal microbiota, resulting in minor but distinguishable differences. Furthermore, different subfertility diagnoses appeared to be associated with differences in microbial profiles. A noteworthy observation was the exclusive presence of *Escherichia coli* (*E. coli*) in both samples of women diagnosed with male factor subfertility. In conclusion, since urinary microbiota profiles seem to represent a diluted version of the vaginal microbiota, vaginal microbiome sampling to predict fertility treatment outcome seems preferable. To enhance the success of fertility treatments, further research is needed to gain deeper insights into a putative causal role of microbiota in the mechanisms of subfertility.

## 1. Introduction

In recent years, the exploration of the human microbiome has emerged as a ground-breaking breakthrough in medical research, shedding light on its profound influence on various aspects of human health. It has been mentioned that microorganisms present in different body parts may have different on-site functions and metabolic activities [1]. The dominant species in the female genital tract are, in general, *Lactobacilli*. Their production of lactic acid, hydrogen peroxide, and bacteriocins serves to prevent the invasion and growth of pathogenic microorganisms. A change in the composition of microbes, due to a variety of causes, can result in a dysbiotic imbalance, which can lead to negative impacts on female fertility and maternal pregnancy outcomes [2].

Within reproductive medicine, associations have been found between microbiota and subfertility [3]. The majority of studies have demonstrated that the dominant species of the urogenital microbiome in patients with lower implantation rates, fewer ongoing pregnancies, and other infertility problems are non-*Lactobacillus* bacteria [4,5,6]. The hypothesis that the chance of achieving a successful pregnancy or pregnancy outcomes can be predicted using the microbiome is supported by several research groups [7,8,9,10,11,12]. Whereas the literature on the association between microbiome and fertility is steadily increasing, and the evidence that a strong link between non-*Lactobacillus* bacteria and subfertility is undeniable, a consensus is still needed regarding which sampling site, (combination of) microbiota, and method of sampling are most representative for fertility outcome prediction.

For example, the female genital tract consists of multiple possible sampling sites such as the fallopian tubes, endometrium, cervix, and vagina [5]. While a sample of the vagina can be collected relatively simply using a swab, the collection of samples from the upper reproductive tract presents some challenges. Transcervical or intraoperatively transfundal endometrial sample collection is prone to contamination due to its invasive technique and the need to involve special devices.

Comparisons have been made between bacteria found in vaginal, cervical, and endometrial samples [13,14,15]. Previous research has suggested that there is extensive overlap between the microorganisms that are present in these anatomical regions [5]. Ekanem et al. showed that the microorganisms from the cervix and vagina are essentially similar [14]. Wee et al. found [13] that the most abundant taxonomic units were the same in cervical and vaginal samples and reported that the relative abundance was generally lower in endometrial microbiota compared to cervical and vaginal samples. On the other hand, Moreno et al. [15] showed differences between endometrial and vaginal microbiota. Some differences were attributed to small differences in percentages per genera, and other differences were attributed to the presence of different genera instead.

As for all predictive tests in development, the applied microbiome sampling technique should be patient-friendly, non-invasive, reproducible, and accurate. Importantly, for the subfertile population, as women often prefer self-collected over clinician-collected vaginal swabs, one study implicated that self-collected vaginal microbial swabs are comparable to those of physician-collected ones [16].

The purpose of the current study was to investigate to what extent the microbial composition of the urogenital tract is comparable to that of the genital tract. More specifically, we investigated if the microbial composition of urine and vaginal samples are comparable. In addition, we examined if different subfertility diagnoses could be associated with differences in microbiota profiles, which could be used in the prediction of pregnancy and determining the subsequent fertility treatment in the future.

A prospective study was conducted in a subfertile population, in which the women self-collected paired mid-stream urine and vaginal samples. The microbial compositions of the collected urinary and vaginal samples were determined using the intergenic spacer profiling (IS-pro) technique [17]. The IS-pro technique offers benefits, including cost-effectiveness, improved time efficiency, and is less labour-intensive compared to the current next-generation sequencing approaches. These characteristics indicate the potential for developing this technique into an on-the-spot diagnostic predictive test, allowing for quick decision-making concerning starting fertility procedures such as IVF or IVF-ICSI.

## 2. Materials and Methods

We performed a prospective study of the urogenital tract microbiota of women with a diagnosis of subfertility and who are scheduled for an in vitro fertilization (IVF) or IVF with intracytoplasmic sperm injection (IVF-ICSI) treatment. According to the World Health Organization (WHO), subfertility is defined as the inability to achieve a clinical pregnancy after twelve months or more of regular, unprotected intercourse. Couples qualifying for IVF treatment have conditions such as previous unsuccessful fertility treatments, tubal dysfunction, endometriosis, and impaired semen quality (a VCM (volume × count × motility) value between 1 and 3 million). Couples qualifying for IVF/IVF-ICSI treatment have a severe male factor, including a VCM cut-off value of less than 1 million. Eight IVF centres in the Netherlands participated. The inclusion criteria for participants were as follows: age between 20 and 44 years with a male partner. The exclusion criteria were as follows: (1) an indication for emergency IVF such as cancer; (2) endometriosis American Fertility Score (AFS) of III/IV and pre-treatment with a gonadotrophin-releasing hormone (GnRH) analogue; (3) use of hormonal contraceptives within 3 months prior to the start of their IVF or IVF-ICSI intake (except women who were using the oral contraceptive pill for the purpose of cycle timing prior to their treatment cycle); and (4) women with a history of any pregnancy including biochemical and clinical miscarriages.

After they were given instructions, the participants were asked to self-collect a ‘clean catch’ mid-stream urine sample and a vaginal swab. The term ‘clean catch’ refers to the practice of cleansing the area around the urethra prior to collecting the urine sample. Furthermore, the participants completed a short questionnaire containing questions about their (recent) use of antibiotics and any urinary tract infections. The materials and procedures are described in detail in the protocol of the ReceptIVFity study [18]. The intergenic spacer profiling (IS-pro) technique was used to analyse the microbiota of both types of samples [17]. The IS-pro technique involves the detection and categorization of the length of the 16S–23S rRNA gene intergenic spacer (IS) region (Figure 1). The length of the IS region is specific for each bacterial species.

The IS regions are amplified using polymerase chain reaction (PCR) with primers that bind to conserved sequences flanking the IS region. This results in the amplification of multiple fragments, each corresponding to different bacterial species present in the sample. The amplified fragments are then fluorescently labelled and separated by capillary electrophoresis. The varying lengths of the IS regions result in distinct peaks, with each peak corresponding to a different bacterial species or strain. The output from the electrophoresis is analysed to generate a profile of the bacterial community in the sample. Each peak in the profile represents a bacterial species, and the height or area of the peak reflects the abundance of that species in the sample (Figure 2) [17].

Species identifications were assigned to peaks based on a validated database compiled of IS-pro fragments obtained from in silico and in vitro IS-pro PCRs of known vagina-associated bacterial species. An internal amplification control (IAC) was used to control the PCR amplification for putative inhibition. A sample passed the quality control when the IAC signal was present in a sufficient amount (3 of 5 IAC peaks > 500 Relative Fluorescence Units (RFUs)) or when a sufficiently high bacterial signal was present (at least one bacterial peak > 20.000 RFU).

This IS-pro technique has been calibrated to the well-known 16S rRNA gene sequencing technique, and it was found that the results were highly comparable and that both can be used to accurately determine the microbiota composition. Nevertheless, the IS-pro analysis has the benefit of rapidity and ease of use. Detailed information about the different steps in the analysis procedure are described in the study protocol of the ReceptIVFity study [18].

The data are reported as the mean (standard deviation) for continuous variables and as a number (percentage) for categorical variables. The normality of continuous variables was assessed using the Shapiro–Wilk W test. For non-normally distributed continuous variables, the Mann–Whitney U test was applied, and for normally distributed continuous variables, t-tests were applied. Fisher’s exact and χ2 tests were used to compare categorical variables between groups. Multiple comparisons of mean ranks for all groups were performed as post hoc tests. Spearman rank order correlation was applied to calculate correlation coefficients. *p* < 0.05 was considered statistically significant. The relative abundance of microbiota per sample was used to perform a correlation clustering of all sample profiles according to the UPGMA method. The relative abundances are given as fluorescence intensity per peak as a percentage of total fluorescence. Next, these data were used to identify the bacterial species using the IS-pro proprietary software suite (inBiome, Amsterdam, The Netherlands, version 0.15.4), and the results are presented as microbial profiles. Cosine correlation was used to compare the abundance of species between samples from both anatomical regions. R2 values were used to show the percentage variation in the microbiota. The statistical analyses were performed using SPSS (Statistical Package for Social Sciences version 25); tornado plots and bar plots were created using Spotfire.

Ethical approval was obtained from the Institutional Medical Ethical Review Board of Erasmus University Medical Centre (METC protocol number 2014-455). Written informed consent was obtained from all participants.

## 3. Results

### 3.1. Study Population

To analyse the difference in microbial composition of the two sampling sites, paired samples of mid-stream urine and vaginal swabs from 303 women were collected prior to start of hormonal fertility treatment. Six women were excluded from analysis; of these, five women had used hormonal contraceptives for more than three weeks prior to start of the IVF/IVF-ICSI treatment and one woman reported a history of miscarriage. In the remaining 297 paired samples, the microbial composition was determined through the use of the IS-pro technique. In 227 of the 297 paired samples, a microbiota profile could be established for both the urine and the vaginal samples. In the remaining 70 paired samples, either only a urine profile (*n* = 5), a vaginal profile (*n* = 60), or neither (*n* = 5) could be established. These 70 samples were all excluded from further analysis (Figure 3).

Table 1 shows the baseline characteristics of the study population. The majority of the women were Caucasian (87%) and had a regular menstrual cycle (77%). Male factor subfertility was the most common cause for the couples to undergo IVF/IVF-ICSI treatment (in 65% of cases). The characteristics of the population included for analysis (n = 227) were separated by subfertility category (male, female, idiopathic, or combination) diagnosis (Table 1). Furthermore, no active urinary tract infection/bacterial vaginosis or use of antibiotics were reported at the time of sample collection. However, seven women indicated that they had been treated with antibiotics for a urinary tract infection and ten women used antibiotics for another indication during the 3 months before inclusion.

### 3.2. Comparison of the Total Urine and Vaginal Bacterial Compositions

First, we analysed the distribution of the various bacterial species that were detected in the urine and vaginal samples (Table 2). In total, 609 bacterial genomes were found in the urinary samples, whereas 795 bacterial genomes were found in the vaginal swabs; these belonged to 19 different bacterial phyla/genera/species. The mean numbers of bacterial species per subject were 2.7 (urinary sample) and 3.5 (vaginal swab). Overall, *Lactobacillus crispatus* and *Lactobacillus iners* were most frequently detected in both types of samples. Significant differences were found in the detection frequencies of *Atopobium vaginae*, *Gardnerella vaginalis*, *Lactobacillus jensenii*, *Megasphaera* sp. type 1, and *Streptococcus oralis*, which were all more frequently detected in the vaginal swabs.

Figure 4 shows the results of the detected bacterial species in the paired urine and vaginal samples from the patients. Small differences between paired urinary and vaginal microbiota profiles were found. However, with the exception of the bacterial species *Sutterella wadsworthensis*, which was only found in one of the two samples in a set of paired samples, all bacterial species were detected in both urinary and vaginal samples.

### 3.3. Comparison of Individual Paired Vaginal and Urine Profiles

Finally, we analysed and compared the paired urine and vaginal profiles for each individual patient. Comparisons of the total microbial urine and vaginal profiles from individual women with the use of cosine correlation yielded an R squared value of 0.78, indicating a high correlation between the urine and vaginal profiles. Furthermore, Figure 5 shows the total (A) and relative (B) abundances of bacteria for both samples of each subject on a horizontal line. In general, dominant species such as *L. crispatus* (blue bars), *L. gasseri* (pink bars), and *L. iners* (yellow bars) were shared between the vaginal and urine profiles of the same individual. The urine profiles of *L. crispatus*-dominated individuals appeared to contain a higher diversity of bacteria than the urine profiles of *L. iners*-dominated individuals.

### 3.4. Microbial Composition in Different Subfertility Diagnosis Categories

Figure 6 illustrates microbiota profiles for each subfertility diagnosis category. Comparing the vaginal profiles for the different subfertility diagnoses, we observed that the bacterial species *E. coli (Escherichia coli)* (*n* = 14) was detected in women with only male factors as the subfertility diagnosis (*n* = 11) or male factors as part of a combined subfertility diagnosis (*n* = 3). The same was observed in the urine microbiota profiles but to a smaller extent (*n* = 9) (women with only male factors as the subfertility diagnosis: *n* = 8; male factors as part of a combined subfertility diagnosis: *n* = 1). Furthermore, the bacterial species *Streptococcus agalactiae* and *Sutterella wadsworthensis* were exclusively found in the urine microbiota profiles of women with male factors as the subfertility diagnosis.

## 4. Discussion

Our results provide evidence that there is a high correlation coefficient (0.78) between the urinary and vaginal microbiota profiles in paired samples of individual patients, and that the small differences in the detected load and presence of bacterial taxa were primarily to the bias against the urine samples (in ~20% of the urine samples, analyses of the bacterial load failed). Furthermore, we showed that subfertility diagnoses may contribute to differences in microbiota composition. Future research will help to clarify the relationship between these differences and clinical outcomes, such as the chance of becoming pregnant.

In this study, we were able to establish more microbiota profiles from vaginal swab samples, and these profiles were generally more informative than those originating from the paired urine samples, i.e., vaginal swabs contained a higher bacterial load and had a higher diversity compared to paired urine samples. We conclude that this makes the vagina the most suitable sampling site for use in daily practice sampling for microbiota profiling. The higher concentration and diversity of bacteria in the vaginal microbiota compared to the urinary microbiota has been previously noted [5,20,21,22]. In addition to a low concentration of bacteria, another explanation for being unable to obtain a microbiota profile could be the high-quality requirements set for passing the quality control (as described in the Methods section).

As a limitation of this study, it should be noted that the women collected the sample themselves. Studies on self-sampling as an alternative to vaginal sampling by a health care professional have shown varying outcomes. Recent studies demonstrated the non-inferiority of self-sampling compared to vaginal sampling by a health care professional [16,23,24]. However, earlier studies suggested a potential impact of the sampling method on the vaginal microbiota [25]. This was mainly due to the fact that earlier studies were based on first-generation molecular techniques and they utilized Sanger sequencing of clone libraries, which can be considered shallow by today’s standards for sequencing. In our study, we performed a technical quality control on all the samples (see Method section) to ensure that the sampling method had no influence on the analysis of the microbiota. In addition, we conducted another study involving a group of more than 200 women, where double samples were taken (self-sampling and sampling by a health care professional). The results showed no differences between the sampling methods [26]. Therefore, the self-collected swab samples used in this study seem to be representative.

Studies describing the female urinary microbiome (urobiome) indicate that the results should be considered as not only the urobiome, but rather as the urogenital microbiome [27], since contamination of the urine sample from the vaginal microbiota is likely to occur during the collection of mid-stream urine. The predominance of the bacterial species *Lactobacillus* in both urinary and vaginal samples in our study was, therefore, not surprising. However, with the present design (clean catch mid-stream urine), we demonstrated that the urinary microbiota is not exclusively a carryover from the vagina, because some bacterial species present in the urine were not found in the vaginal samples from the same woman, and vice versa. It is expected that the use of first-voided urine collection would, most likely, have led to a greater resemblance to the vaginal sample as a result of contamination [28].

We observed significant differences between paired samples in terms of specific species, including an increased detection of *Atopobium vaginae*, *Gardnerella vaginalis*, *Lactobacillus jensenii*, *Megasphaera* sp.* type 1*, and *Streptococcus oralis*, which were all more frequently detected in the vaginal swabs. *Lactobacillus jensenii* is one of the dominant bacteria in a healthy vaginal microbiota and has been correlated with better fertility outcomes. In contrast, the presence of *Atopobium vaginae* and *Gardnerella vaginalis* is often associated with bacterial vaginosis, which is linked to an increased risk of premature labour, reproductive tract infections, and potential negative impacts on fertility outcomes [29,30,31]. Given the impact of these species on fertility, the choice of which sample type to analyse (urine or vaginal) for adequate determination of the microbiota could become important in the future to better predict the chances of pregnancy in subfertile patients.

Recent studies, including those from our own group, indicate that the composition, e.g., the presence or absence of different bacterial species, may be linked to pregnancy outcomes after fertility treatments [12,28,29,30,31,32,33,34,35,36,37,38] These studies found associations between detected bacterial species and the occurrence of compositional perturbations in the microbiome of the urogenital tract, which could function as markers for the presence of an optimal environment for embryo implantation.

In the current study, we also examined whether different subfertility diagnoses led to different microbiota profiles. An exclusive presence of *E. coli* was demonstrated in both urine and vaginal samples of women with subfertility due to a male factor diagnosis. We speculate that the *E. coli* found in women with male factor as subfertility diagnosis initially originates from their male partner and becomes part of the seminovaginal microbiome. *E. coli* is regarded as one of the major pathogens of postcoital urinary tract infections in women [39]. To confirm the male origin of the *E. coli*, a microbiota profile of the semen should, in future research, be analysed simultaneously with the vaginal microbiota profile.

The fact that the bacterial species *E. coli* was found exclusively in the samples (both urine and vaginal) from women with subfertility due to a male factor diagnosis could suggest that this species may also play a role in male fertility. Bacterial infections are considered a substantial cause of male subfertility [40]. Indeed, it has been reported that bacteriospermia was directly related to 15% of male subfertility in a group of men treated with ART [41]. Moreover, Michel et al. [42] showed that *E. coli* was a frequent pathogen causing epididymitis and was also involved in other causes of impaired male fertility. The molecular mechanism by which fertility is affected is complex and multifactorial. The suggested pathogenesis of this cause of subfertility is that it can lead to fibrosis and thereby to obstruction of the epididymis. Another mechanism might be that *E. coli* directly exerts its detrimental effect on spermatozoa by rapidly adhering to spermatozoa, causing morphological alterations and leading to the immobilization of the spermatozoa [40], impairment of the acrosome reaction [43], and sperm damage due to bacterial production of toxins and metabolites [44]. *E. coli* adherence to spermatozoa and its presence within the seminal fluid would explain the transmission from the male to the female environment.

It is also known that sexual intercourse without a condom has no effect on vaginal *Lactobacilli* but it does lead to elevated levels of *E. coli* [45]. Indeed, all the women in our study were trying to conceive and, therefore, did not use condoms, which might explain the presence of *E. coli* species in their vaginal microbiome, and is a sign that the vaginal microbiome could be affected by sexual intercourse/male genital tract microbiota [46]. However, Eschenbach et al. [47] showed that this effect was also seen in urine; they noted increased counts of coliforms in both the vagina and urine after sexual intercourse. However, the contribution of the sperm microbiome to the female microbiome is minimal compared to the abundance of the vaginal microbiome, so the overall makeup will not be significantly altered by the sperm microbiome.

The presence of *Streptococcus agalactiae* and *Sutterella wadsworthensis* were only detected in the urine samples of the male factor subfertility group. While Streptococcus agalactiae has been linked to preterm birth, there is currently no direct evidence linking this species to subfertility.

## 5. Conclusions

Our study has provided important insights into the similarities and differences in the microbiota profiles of urine and vaginal samples in a subfertile population. This work adds to the growing body of literature suggesting that the urinary microbiota mainly represents a diluted version of the vaginal microbiota. However, the differences highlight the importance of adequate sampling site selection and collection for accurate microbiota profiling. Notably, *Escherichia coli* was exclusively detected in both urine and vaginal samples of women diagnosed with male factor subfertility, suggesting a possible role of *E. coli* in subfertility that warrants further investigation. To increase the success rates of fertility treatments, future research should focus on elucidating the precise role of microbiota in subfertility and exploring the implications of microbiota on reproductive health. Based on our study, we would recommend vaginal samples for this follow-up study.

## 6. Patents

Paul Savelkoul, Jonathan de Jonge, and Dries Budding hold the patents ‘Microbial Population Analysis’ (9506109) and ‘Microbial Population Analysis’ (20170159108), both licensed to ARTPred B.V.

Jonathan de Jonge, Dries Budding, and Martine Bos have written patent applications titled ‘Method and Kit for Predicting the Outcome of an Assisted Reproductive Technology Procedure’ (392EPP0) and ‘Method and Kit for Altering the Outcome of an Assisted Reproductive Technology Procedure’ for ARTPred.

Joop Laven is a co-applicant on an Erasmus MC patent (‘New Method and Kit for Prediction of Success of In Vitro Fertilization’) licensed to ARTPred B.V.

## Figures and Tables

**Figure 1 microorganisms-12-01789-f001:**
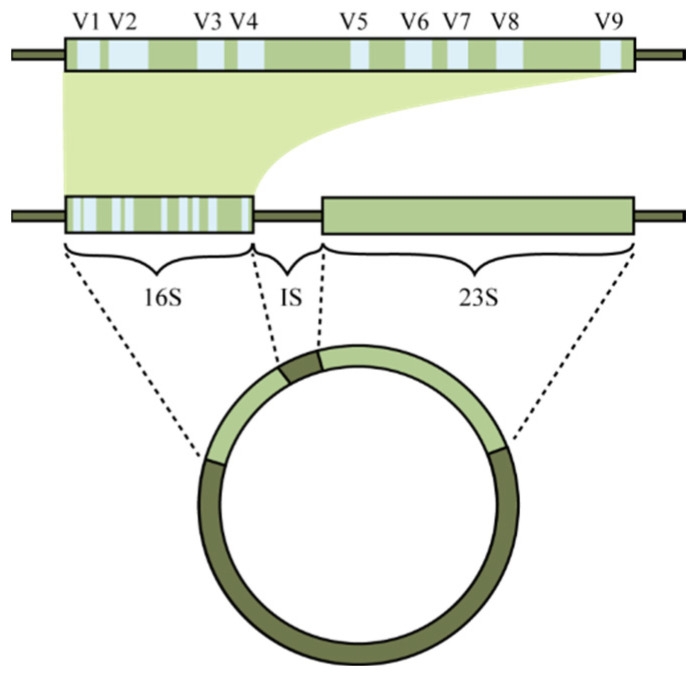
A representation of the circular chromosome of bacteria. The 16S and 23S ribosomal RNA genes are highlighted together with the intergenic space (IS) region. V1–V9 marks the variable regions, with the conserved region between them. (Reprinted from Koedooder et al. [5] with permission).

**Figure 2 microorganisms-12-01789-f002:**
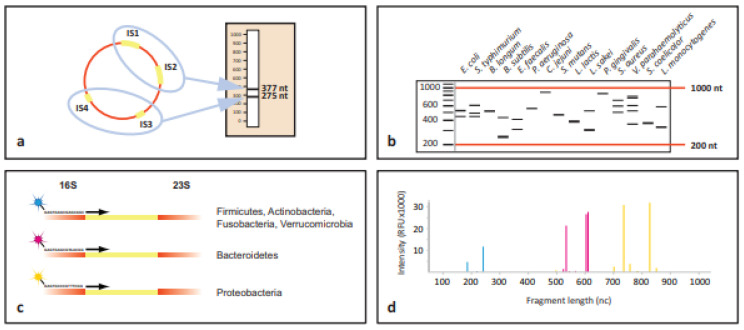
Schematic representation of the concept behind IS-pro procedure. (**a**) All bacterial species contain at least one IS region in their chromosome. However, many species contain multiple alleles of the IS region. These regions may vary between different alleles. Depicted here is the situation for Enterococcus faecalis, which contains four alleles of the IS region. Two have a length of 275 nucleotides (nc), and the other two have a length of 377 nc. When amplified, a profile specific for *E. faecalis* is obtained. (**b**) IS profiles are highly diverse between different species. The fact that species commonly have multiple alleles with different lengths dramatically increases the differentiation potential between species. (**c**) By amplifying IS fragments using phylum-specific fluorescently labelled primers, another layer of information is added. (**d**) When an IS profile is made for a sample containing multiple species from different phyla, peaks with different lengths, heights, and colours are found. These correspond to the species, abundance, and phylum. A peak profile may be translated into a list of bacterial species by a software algorithm linked to a database of IS profiles of known bacterial species. (Reprinted from Budding et al. [19] with permission).

**Figure 3 microorganisms-12-01789-f003:**
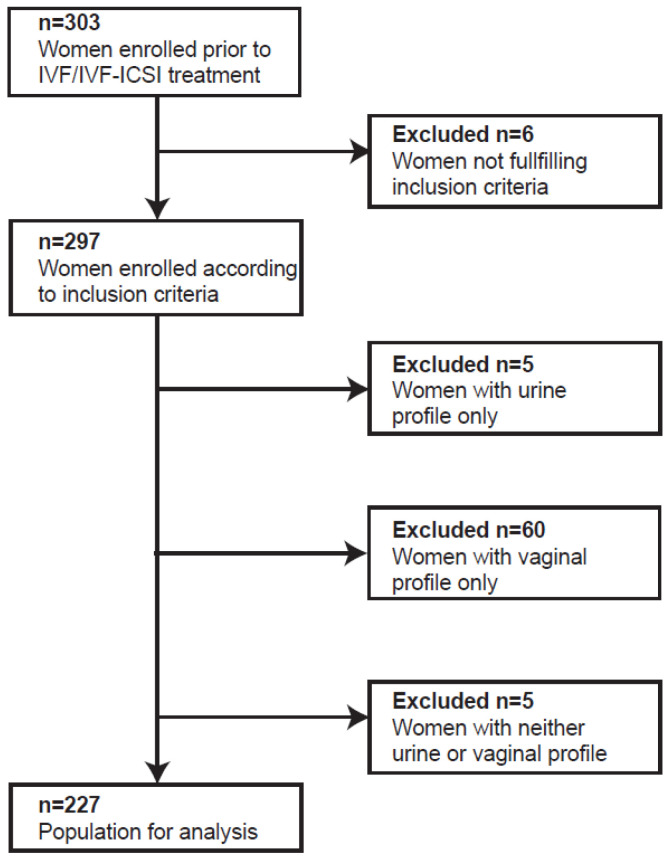
Flow chart of study population.

**Figure 4 microorganisms-12-01789-f004:**
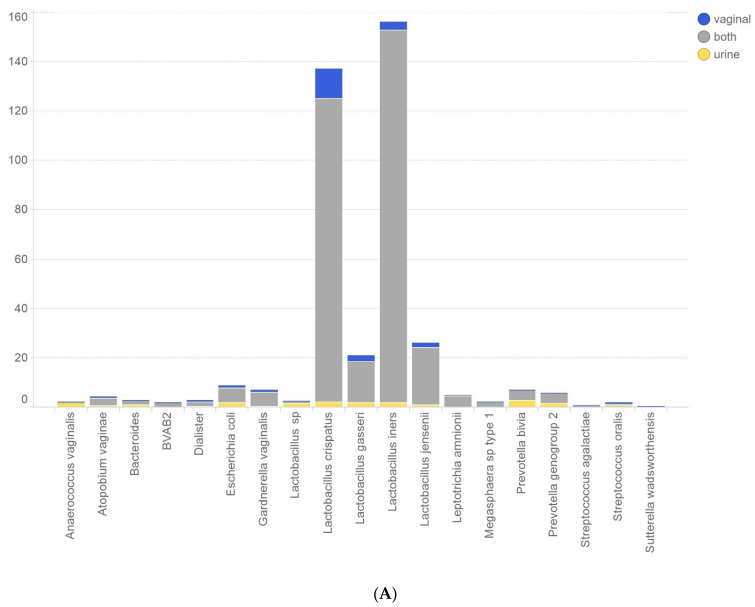
Bar graph showing the bacterial species distribution over the paired samples. Blue: bacterial species was only found in the vaginal sample; grey: bacterial species was found in both of the paired samples; yellow: bacterial species was only found in the urine sample. Number of subjects (**A**) and relative distribution of bacterial species (**B**) are indicated on the *Y*-axis.

**Figure 5 microorganisms-12-01789-f005:**
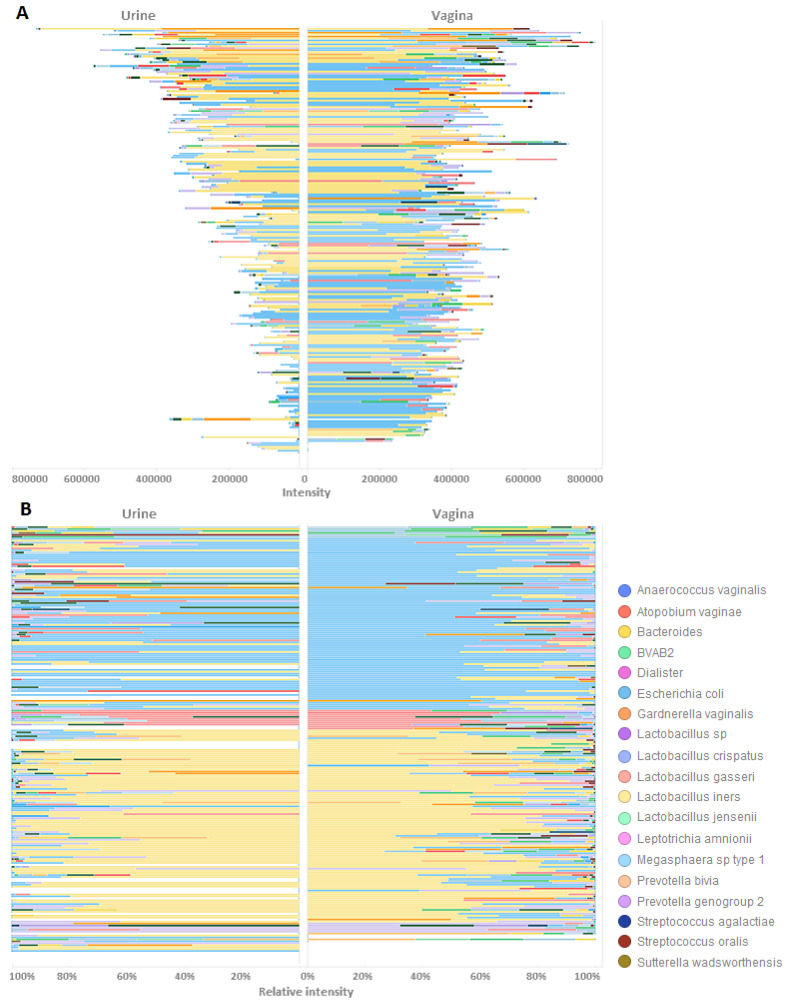
Tornado plots where each horizontal line represents one subject, with urine profiles on the left and vaginal profiles on the right of the vertical axis. White bars indicate samples that failed to yield data during the analysis of one sample type. (**A**) The IS-pro signal intensity distribution for each sample showing the identified taxa by colour. (**B**) The relative abundance of species found using IS-pro in each sample. Legend clarifying colours for taxa found using IS-pro.

**Figure 6 microorganisms-12-01789-f006:**
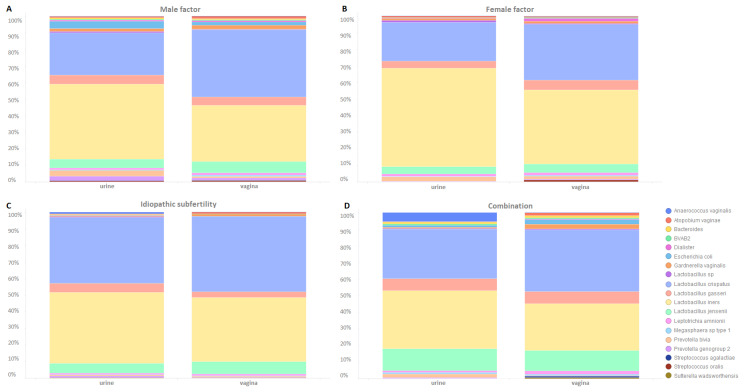
(**A**–**D**) Bar plots of microbiota profiles for each subfertility diagnosis category and sample type (urine, vaginal).

**Table 1 microorganisms-12-01789-t001:** Characteristics of study population and subgroups of subfertility diagnosis.

	Total Study Population n = 227	Male Factor n = 147	Female Factor n = 21	Idiopathic Subfertility n = 34	Combination * n = 21
Age (years) ^a^	31.64 (4.44)	31.09 (4.41)	31.61 (3.13)	34.10 (3.83) ^1^	31.01 (4.32)
Ethnicity ^b^					
Caucasian	197 (86.8)	132 (89.8)	16 (76.2)	28 (82.4)	17 (81.0)
Non-Caucasian	22 (9.7)	11 (7.5)	3 (14.3)	5 (14.7)	3 (14.3)
Body mass index (kg/m^2^) ^a^	24.55 (4.47)	24.67 (4.23)	25.21 (6.33)	22.54 (3.50)	25.60 (4.53)
Use of medication ^b^					
Yes	54 (23.8)	31 (21.1)	4 (19.0)	7 (20.6)	10 (47.6)
No	171 (75.3)	115 (78.2)	17 (81.0)	26 (76.5)	11 (52.4)
Menstrual cycle ^b^					
Regular	174 (76.7)	119 (81.0)	17 (81.0)	26 (76.5)	7 (33.3)
Mostly regular	21 (9.3)	14 (9.5)	1 (4.8)	6 (17.6)	0 (0.0)
Irregular	26 (11.5)	11 (7.5)	3 (14.3)	0 (0.0)	11 (52.4)
Absent	2 (0.9)	1 (0.7)	0 (0.0)	0 (0.0)	1 (4.8)
Duration of subfertility (years) ^a^	2.71 (1.88)	2.51 (2.00)	3.13 (1.56)	3.48 (1.73) ^2^	2.42 (1.34)

Data are presented as the ^a^ mean (standard deviation) or a ^b^ number (percentage). * More than one cause for subfertility diagnosis was found. An ANOVA test with post hoc Bonferroni was performed to test differences; ^1^ male vs. idiopathic, *p* < 0.05; ^2^ male vs. idiopathic, *p* < 0.05.

**Table 2 microorganisms-12-01789-t002:** Overall overview of bacterial species detected in urinary and vaginal samples.

Bacterial Phyla/Genera/Species	Urinary Sample N = 227 (%)	Vaginal Swab N = 227 (%)	*p*-Value
*Anaerococcus vaginalis*	10 (4.4)	12 (5.3)	0.662 ^a^
*Atopobium vaginae*	18 (7.9)	51 (22.5)	0.000016 ^a^
*Bacteroides*	18 (7.9)	18 (7.9)	1.000 ^a^
*BVAB2*	13 (5.7)	22 (9.7)	0.113 ^a^
*Dialister*	22 (9.7)	25 (11.0)	0.644 ^a^
*Escherichia coli*	9 (4.0)	14 (6.2)	0.285 ^a^
*Gardnerella vaginalis*	27 (11.9)	50 (22.0)	0.004 ^a^
*Lactobacillus* sp.	9 (4.0)	5 (2.2)	0.278 ^a^
*Lactobacillus crispatus*	138 (60.8)	156 (68.7)	0.077 ^a^
*Lactobacillus gasseri*	24 (10.6)	36 (15.9)	0.096 ^a^
*Lactobacillus iners*	136 (59.9)	140 (61.7)	0.701 ^a^
*Lactobacillus jensenii*	62 (27.3)	85 (37.4)	0.021 ^a^
*Leptotrichia amnionii*	12 (5.3)	19 (8.4)	0.193 ^a^
*Megasphaera* sp. *type 1*	17 (7.5)	32 (14.1)	0.005 ^a^
*Prevotella bivia*	34 (15.0)	31 (13.7)	0.688 ^a^
*Prevotella genogroup 2*	45 (19.8)	35 (15.4)	0.218 ^a^
*Streptococcus agalactiae*	2 (0.9)	5 (2.2)	0.449 ^b^
*Streptococcus oralis*	11 (4.8)	54 (23.8)	8.3178 × 10^−9 a^
*Sutterella wadsworthensis*	2 (0.9)	5 (2.2)	0.253 ^b^

Data are presented as a number (percentage). ^a^ Chi-square test, ^b^ Fisher’s exact test.

## Data Availability

The database of the IS-pro findings of this study is available from InBiome B.V. but restrictions apply to the availability of these data, which were used under a license for the current study, and so are not publicly available. Data are however available from the authors upon reasonable request and with permission from InBiome B.V.

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
