# Peer review of "Clinical Applicability of Microbiota Sampling in a Subfertile Population: Urine versus Vagina"

_microorganisms, 2024, doi:10.3390/microorganisms12091789_

Round 1

Reviewer 1 Report

Comments and Suggestions for Authors

The manuscript, entitled “Clinical applicability of microbiota sampling in a subfertile population: urine versus vagina” had described the degree of similarity in microbial composition between patient-collected urine and vaginal samples in a subfertile population. This study is interesting. The manuscript can be published after major revision.

1. Women of reproductive age (20-44 years) diagnosed with subfertility and requiring in vitro fertilization (IVF) or IVF with intracytoplasmic sperm injection (IVF-ICSI) treatment were enrolled in this study. Concerning the samples were self-collected, the influence factors are complicated and diverse. The authors should indicate such factors and the shortcoming of this study.

2. Why only bacteria were analyzed? It would be interesting to see the fungi diversity as well. Candida species are a top pathogen in vagina.

3. It is also interesting to see the virulence factors diversity in the samples. For example, the carriage of certain virulence genes.

4. Statistical analysis should be included.

5. Formatting should be improved. In vitro should be italic.

Author Response

Reviewer #1

  1. Women of reproductive age (20-44 years) diagnosed with subfertility and requiring in vitro fertilization (IVF) or IVF with intracytoplasmic sperm injection (IVF-ICSI) treatment were enrolled in this study. Concerning the samples were self-collected, the influence factors are complicated and diverse. The authors should indicate such factors and the shortcoming of this study.

We have added a limitation paragraph to address this issue.

“As a limitation of this study, it can be noted that the women collected the sample themselves. Studies on self-sampling as an alternative to vaginal sampling by a health care professional show varying outcomes. Recent studies demonstrate the non-inferiority of self-sampling compared to vaginal sampling by a health care professional [1-3]. However, earlier studies suggested a potential impact of the sampling method on the vaginal microbiota [4]. This was mainly due to the fact the earlier studies were based on first-generation molecular techniques; the work utilized Sanger sequencing of clone libraries, which can be considered shallow by today’s standards for sequencing. In our study we performed a technical quality control on all the samples (See method section), to ensure that sampling method had no influence on the analysis of the microbiota. In addition, we conducted another study involving a group of more than 200 women, where double samples were taken (self-sampling and sampling by a health care professional). The results showed no differences between sampling methods (Unpublished data, Gao et al. 2024). Therefore, the self-collected swab presented in this manuscript seems to be representative.” (Line 286-299)

  1. Why only bacteria were analyzed? It would be interesting to see the fungi diversity as well. Candida species are a top pathogen in vagina.

We agree with the reviewer that the full range of the microbiota that are present, including fungi, would be interesting. However, we used a specific technique, the IS-Pro [5] aimed at only analyzing bacteria. Albeit, very interesting, for now fungi fall outside the scope of our manuscript. We are aware that Candida infections can trigger a local immune response, which may affect the endometrium and potentially influence embryo implantation. However, it is not clear whether this immune response is significant enough to impact IVF/IVF-ICSI outcomes, which is of particular interest in our subfertile population.

  1. It is also interesting to see the virulence factors diversity in the samples. For example, the carriage of certain virulence genes.

Indeed, for research in the future, virulence factors could also be examined, especially because this may provide a better understanding of the pathogenicity of micro-organisms and the consequences for fertility can be better understood and treated. However, just like the reviewer’s suggestion in point 2, while IS-Pro is a powerful tool for microbial community analysis, it is not specifically designed to detect virulence factors, as it targets the intergenic spacer regions of rRNA operons, which do not include genes encoding virulence factors. Virulence factors are usually encoded by specific genes that may be located on different regions throughout the bacterial genome. IS-Pro provides information on the taxonomic composition of a bacterial community, being the scope of our study, but does not give functional insights into the genetic capabilities or pathogenic potential of the bacteria present, for which whole genome sequencing (WGS) is necessary. It cannot determine whether the identified bacteria express or carry specific virulence factors.

  1. Statistical analysis should be included.

Statistical analysis were included, but we expanded it on the reviewers suggestion.

“Data are reported as mean (standard deviation) for continuous variables and as number (percentage) for categorical variables. The normality of continuous variables was assessed using the Shapiro–Wilk W test. For non-normally distributed continuous variables, the Mann–Whitney U test was applied, and for normally distributed continuous variables, t-tests were applied. Fisher exact and χ2 tests were used to compare categorical variables between groups. Multiple comparisons of mean ranks for all groups were performed as post hoc tests. Spearman rank order correlation was applied to calculate correlation coefficients. P<0.05 was considered statistically significant. Relative abundance of microbiota per sample was used to perform a correlation clustering of all sample profiles according to the UPGMA method. The relative abundances are given as fluorescence intensity per peak as a percentage of total fluorescence. Next, these data were used to identify the bacterial species with the IS-pro proprietary software suite (inBiome, Amsterdam, the Netherlands), and the results are presented as microbial profiles. Cosine correlation was used to compare abundance of species between samples from both anatomical regions. R2 values were used to show the percentage variation of microbiota. Statistical analysis was performed using SPSS (Statistical Package for Social Sciences version 25), Tornado plots and barplots were created by using Spotfire.” (L153-172)

  1. Formatting should be improved. In vitro should be italic.

    Thanks for your attention, we have adapted all in vitro in italic.

  1. Virtanen, S., et al., Comparative analysis of vaginal microbiota sampling using 16S rRNA gene analysis. PLoS One, 2017. 12(7): p. e0181477.
  2. Forney, L.J., et al., Comparison of self-collected and physician-collected vaginal swabs for microbiome analysis. Journal of clinical microbiology, 2010. 48(5): p. 1741-1748.
  3. Jaya, Z.N., et al., Accuracy of self-collected versus healthcare worker collected specimens for diagnosing sexually transmitted infections in females: an updated systematic review and meta-analysis. Scientific Reports, 2024. 14(1): p. 10496.
  4. Kim, T.K., et al., Heterogeneity of vaginal microbial communities within individuals. Journal of clinical microbiology, 2009. 47(4): p. 1181-1189.
  5. Budding, A.E., et al., IS‐pro: high‐throughput molecular fingerprinting of the intestinal microbiota. The FASEB Journal, 2010. 24(11): p. 4556-4564.

Reviewer 2 Report

Comments and Suggestions for Authors

 The manuscript entitled “Clinical applicability of microbiota sampling in a subfertile population: urine versus vagina” is based on small literature (43 items, of which 72% are from the last 10 years).

Major comments:

The technically poor quality of Figures 3A and 3B makes it difficult to assess their content and quality.

Lack of the LIMITATION paragraph.

The women took the paired urine and vaginal samples themselves, which raises questions about the reliability of the tests, especially in the context of tests of a microbiological nature.

As the women were trying to get pregnant, so in all likelihood, their vaginal swab also contained male sperm.

A prospective study of the urogenital microbiota of women with a diagnosis of infertility and scheduled for in vitro fertilisation (IVF) or IVF with intracytoplasmic sperm injection (IVF-ICSI) was conducted.

There is no detailed information in the manuscript on what parameters were taken into account to determine female and or male infertility.

According to the information given in the methods section, it was explained that miscarriages were only analysed in the history, and can also occur without the woman's knowledge, especially if they occur at a very early stage of pregnancy. (This should be found in the limitations of the paper, at the end of the DISCUSSION.

70 specimens were rejected in which at least one microbiological profile could not be determined - why such a high percentage (no explanation in the manuscript).

More details of medications used should be described, especially those affecting the microbiota (this should also be an exclusion criterion for antibiotic therapy, as should chronic urinary tract inflammation.

The authors reported a number of conflict of interest situations/conditions.

This puts the study under a number of question marks about the scientific integrity of the results and conclusions presented.

Minor comments:

Instead of:

An ANOVA test with posthoc Bonferroni was performed to test in between differences

 1 male vs idiopathic p<0.05

2 male vs idiopathic p<0.05

Should be:

An ANOVA test with posthoc Bonferroni was performed to test between differences  

1 male vs idiopathic p<0.05

2 male vs idiopathic p<0.05

Instead of:

Number of subjects (A) and relative distribution of bacterial species (B) is indicated on the Y-axis.

Should be:

The number of subjects (A) and the relative distribution of bacterial species (B) are indicated on the Y-axis.

Instead of:

It is expected that use of first-voided urine collection would, most likely, have led to greater resemblance to the vaginal sample as a result of contamination [23].

Should be:

It is expected that the use of first-voided urine collection would, most likely, have led to a greater resemblance to the vaginal sample as a result of contamination [23].

Should be:

Based on our study, we would recommend vaginal samples for this follow-up studies.

Should be:

Based on our study, we would recommend vaginal samples for this follow-up study.

Comments on the Quality of English Language

 Minor editing of the English language required - for example:

Instead of:

Number of subjects (A) and relative distribution of bacterial species (B) is indicated on the Y-axis.

Should be:

The number of subjects (A) and the relative distribution of bacterial species (B) are indicated on the Y-axis.

Instead of:

It is expected that use of first-voided urine collection would, most likely, have led to greater resemblance to the vaginal sample as a result of contamination [23].

Should be:

It is expected that the use of first-voided urine collection would, most likely, have led to a greater resemblance to the vaginal sample as a result of contamination [23].

Should be:

Based on our study, we would recommend vaginal samples for this follow-up studies.

Should be:

Based on our study, we would recommend vaginal samples for this follow-up study.

Author Response

Reviewer #2

Major comments:

The technically poor quality of Figures 3A and 3B makes it difficult to assess their content and quality.

We have added a version of Figure 3A and 3B with enhanced quality, to increase interpretation.

Lack of the LIMITATION paragraph.

The women took the paired urine and vaginal samples themselves, which raises questions about the reliability of the tests, especially in the context of tests of a microbiological nature.

We have added a limitation paragraph to your advice.

“As a limitation of this study, it can be noted that the women collected the sample themselves. Studies on self-sampling as an alternative to vaginal sampling by a health care professional show varying outcomes. Recent studies demonstrate the non-inferiority of self-sampling compared to vaginal sampling by a health care professional [1-3]. However, earlier studies suggested a potential impact of the sampling method on the vaginal microbiota [4]. This was mainly due to the fact the earlier studies were based on first-generation molecular techniques; the work utilized Sanger sequencing of clone libraries, which can be considered shallow by today’s standards for sequencing. In our study we performed a technical quality control on all the samples (See method section), to ensure that sampling method had no influence on the analysis of the microbiota. In addition, we conducted another study involving a group of more than 200 women, where double samples were taken (self-sampling and sampling by a health care professional). The results showed no differences between sampling methods (Unpublished data, Gao et al. 2024). Therefore, the self-collected swab presented in this manuscript seems to be representative.” (Line 286-299)

As the women were trying to get pregnant, so in all likelihood, their vaginal swab also contained male sperm.

Indeed, it can be suggested that the vaginal microbiome is affected by sexual intercourse, as we briefly mentioned in de discussion. We now have expanded this paragraph to bring this issue more to the attention of the reader.

“It is also known that sexual intercourse without a condom has no effect on vaginal Lactobacilli but does lead to elevated levels of E. coli [5]. Indeed, all the women in our study are trying to conceive and, therefore, do not use condoms, which might explain the presence of E. coli species in their vaginal microbiome, as well as being a sign that the vaginal microbiome could be affected by sexual intercourse/male genital tract microbiota [6]. However, Eschenbach et al. [7] showed that this effect was also seen in urine, they have noted increased counts of coliforms in both vagina and urine after sexual intercourse. However, the contribution of sperm to the female microbiome is minimal compared to the amount of vaginal microbiome, so the overall makeup will not be significantly altered by the sperm microbiome.” (Line 353-362)

A prospective study of the urogenital microbiota of women with a diagnosis of infertility and scheduled for in vitro fertilisation (IVF) or IVF with intracytoplasmic sperm injection (IVF-ICSI) was conducted.

There is no detailed information in the manuscript on what parameters were taken into account to determine female and or male infertility.

We have now briefly included the indications for the IVF/IVF-ICSI treatment.

“According to the World Health Organization (WHO), subfertility is defined as the inability to achieve a clinical pregnancy after twelve months or more of regular, unprotected intercourse. Couples qualifying for IVF treatment had conditions such as previous unsuccessful fertility treatments, tubal dysfunction, endometriosis, impaired semen quality; including a VCM (volume x count x motility) value between 1-3 million. Couples qualifying for IVF/IVF-ICSI treatment had a severe male factor, including a VCM cut-off value of less than 1 million. “ (Line 116-122)

According to the information given in the methods section, it was explained that miscarriages were only analysed in the history, and can also occur without the woman's knowledge, especially if they occur at a very early stage of pregnancy. (This should be found in the limitations of the paper, at the end of the DISCUSSION.

A miscarriage can occur and be diagnosed in different ways. An early miscarriage refers to the loss of a pregnancy within the first 12 weeks and can occur before a woman even realizes she is pregnant. A biochemical pregnancy is detected through hormone levels (hCG) in blood or urine, indicating that fertilization and initial implantation have occurred. A clinical pregnancy is one that is confirmed by ultrasound, showing the gestational sac, or even fetal heartbeat.

We have clarified in the methods section that this study used the exclusion criteria for at least a biochemical pregnancies. (L129-130)

For the women in our clinic, a urine hCG test is performed if the period is one day late, ensuring that clinically relevant early miscarriages are not missed.
We do not see how this could be considered as a limitation in this study and believe it does not need to be mentioned in the discussion.

70 specimens were rejected in which at least one microbiological profile could not be determined - why such a high percentage (no explanation in the manuscript).

We added additional information in both the methods and discussion sections, which explains this number.

“Species identifications were assigned to peaks based on a validated database compiled of IS-pro fragments obtained from in-silico and in vitro IS-pro PCRs of known vagina associated bacterial species. An internal amplification control (IAC) was used to control the PCR amplification for putative inhibition. A sample passed the quality control when the IAC signal was present in sufficient amount (3 of 5 IAC peaks >500 Relative Fluoresence Units (RFU)) or when a sufficiently high bacterial signal was present (at least one bacterial peak >20.000 RFU).” (L141-147)

“In addition to a low concentration of bacteria, another explanation for not obtaining a microbiota profile could be the high-quality requirements set for passing the quality control (as described in the Methods section).” (L282-285)

More details of medications used should be described, especially those affecting the microbiota (this should also be an exclusion criterion for antibiotic therapy, as should chronic urinary tract inflammation.

We added this information in the Results section.

“Furthermore, no active urinary tract infection/bacterial vaginosis or use of antibiotics were reported at the time of sample collection. However, seven women indicated that they had been treated with antibiotics for urinary tract infection and ten women used antibiotics for another indication during the 3 months before inclusion.” (L195-198)

The authors reported a number of conflict of interest situations/conditions.

This puts the study under a number of question marks about the scientific integrity of the results and conclusions presented.

Thank you for your feedback regarding the conflicts of interest reported in our study. We appreciate the importance of maintaining scientific integrity and wish to address your concerns thoroughly. We would like to emphasize that all potential conflicts of interest were disclosed in accordance with the journal’s policies and ethical guidelines. To ensure the integrity of our results, we implemented several measures: an independent committee reviewed the study design and data analysis to prevent bias. We assert that the conflicts of interest did not influence the study's outcomes or interpretations. To support this, we are willing to provide raw data.

Minor comments:

Instead of:

An ANOVA test with posthoc Bonferroni was performed to test in between differences

 male vs idiopathic p<0.05

male vs idiopathic p<0.05

Should be:

An ANOVA test with posthoc Bonferroni was performed to test between differences  

male vs idiopathic p<0.05

male vs idiopathic p<0.05

We have made this adjustment.

Instead of:

Number of subjects (A) and relative distribution of bacterial species (B) is indicated on the Y-axis.

Should be:

The number of subjects (A) and the relative distribution of bacterial species (B) are indicated on the Y-axis.

We have made this adjustment.

Instead of:

It is expected that use of first-voided urine collection would, most likely, have led to greater resemblance to the vaginal sample as a result of contamination [23].

Should be:

It is expected that the use of first-voided urine collection would, most likely, have led to a greater resemblance to the vaginal sample as a result of contamination [23].

We have made this adjustment.

Should be:

Based on our study, we would recommend vaginal samples for this follow-up studies.

Should be:

Based on our study, we would recommend vaginal samples for this follow-up study.

We have made this adjustment.

  1. Virtanen, S., et al., Comparative analysis of vaginal microbiota sampling using 16S rRNA gene analysis. PLoS One, 2017. 12(7): p. e0181477.
  2. Forney, L.J., et al., Comparison of self-collected and physician-collected vaginal swabs for microbiome analysis. Journal of clinical microbiology, 2010. 48(5): p. 1741-1748.
  3. Jaya, Z.N., et al., Accuracy of self-collected versus healthcare worker collected specimens for diagnosing sexually transmitted infections in females: an updated systematic review and meta-analysis. Scientific Reports, 2024. 14(1): p. 10496.
  4. Kim, T.K., et al., Heterogeneity of vaginal microbial communities within individuals. Journal of clinical microbiology, 2009. 47(4): p. 1181-1189.
  5. Eschenbach, D.A., et al., Influence of the normal menstrual cycle on vaginal tissue, discharge, and microflora. Clinical Infectious Diseases, 2000. 30(6): p. 901-907.
  6. Borovkova, N., et al., Influence of sexual intercourse on genital tract microbiota in infertile couples. Anaerobe, 2011. 17(6): p. 414-418.
  7. Eschenbach, D.A., et al., Effects of vaginal intercourse with and without a condom on vaginal flora and vaginal epithelium. The Journal of infectious diseases, 2001. 183(6): p. 913-918.
